# Predicting Male Infertility Using Artificial Neural Networks: A Review of the Literature

**DOI:** 10.3390/healthcare12070781

**Published:** 2024-04-03

**Authors:** Vivian Schmeis Arroyo, Marco Iosa, Gabriella Antonucci, Daniela De Bartolo

**Affiliations:** 1Department of Psychology, University Sapienza of Rome, 00185 Rome, Italymarco.iosa@uniroma1.it (M.I.); gabriella.antonucci@uniroma1.it (G.A.); 2Santa Lucia Foundation, Scientific Institute for Research, Hospitalization and Health Care (IRCCS), 00179 Rome, Italy

**Keywords:** machine learning, male infertility, artificial intelligence, statistical models

## Abstract

Male infertility is a relevant public health problem, but there is no systematic review of the different machine learning (ML) models and their accuracy so far. The present review aims to comprehensively investigate the use of ML algorithms in predicting male infertility, thus reporting the accuracy of the used models in the prediction of male infertility as a primary outcome. Particular attention will be paid to the use of artificial neural networks (ANNs). A comprehensive literature search was conducted in PubMed, Scopus, and Science Direct between 15 July and 23 October 2023, conducted under the Preferred Reporting Items for Systematic Reviews and Meta-Analyses (PRISMA) guidelines. We performed a quality assessment of the included studies using the recommended tools suggested for the type of study design adopted. We also made a screening of the Risk of Bias (RoB) associated with the included studies. Thus, 43 relevant publications were included in this review, for a total of 40 different ML models detected. The studies included reported a good quality, even if RoB was not always good for all the types of studies. The included studies reported a median accuracy of 88% in predicting male infertility using ML models. We found only seven studies using ANN models for male infertility prediction, reporting a median accuracy of 84%.

## 1. Introduction

Demographers tend to interpret (define) infertility as childlessness in a population of women of reproductive age, whereas the epidemiological definition refers to trying for or time to pregnancy in a population of women exposed to the probability of conception [1]. Infertility is a universal health issue, covering about 8 to 12% of couples worldwide [2], and as a chronic condition, it can result in distress, stigma, and financial hardship, affecting people’s mental and psychosocial well-being [3]. 

Approximately 30% of infertility cases are attributed to male factors [4]. Recent meta-analysis studies show that male factors are present in 20–70 percent of infertility cases [5]. These findings are significantly broader than previously reported [4,5]. However, the wide range of male infertility in meta-analysis studies [6] may not reflect the prevalence of this complication in all parts of the world because of reasons, such as the lack of rigorous statistical methods that include bias, heterogeneity in data collection, and cultural constraints [6]. Indeed, the inability to conceive a child can be emotionally taxing, leading to feelings of inadequacy, and men may contemplate their masculinity, as societal standards frequently associate potency with manliness. The constant demand to excel and procreate can intensify these feelings [7]. As mentioned before, in almost all cases, it is not only the man who suffers from it, as couples also facing fertility issues often experience increased tension in their relationships. The strain comes from various factors, such as the emotional reaction to fertility treatments, the financial burden, and the pressure to conceive [8]. Furthermore, the necessity for timed sexual intercourses and a heightened focus on fertility can take away the spontaneous nature of such interactions, causing them to feel more like clinical procedures than intimate moments, which may lead to an environment unsuitable for conception [9].

Recent studies addressed the investigation of genetic abnormalities [10], such as chromosomal abnormalities and gene mutations [11], as factors affecting sperm production and function [10]. Furthermore, the role of hormonal imbalances, such as low levels of testosterone or high levels of prolactin, in disrupting the normal functioning of the reproductive system and sperm production has been investigated [12]. Indeed, lifestyle factors, including smoking, excessive alcohol consumption, drug use, and exposure to environmental toxins, can also contribute to male infertility [13,14]. Recently, there has been an increasing interest in chemical exposure as a cause of male infertility [14], which likely plays a substantial role in the sperm count trend. 

Last but not least, risk factors exhibit a correlation with obesity, which is a known risk factor for diminished-quality sperm [15]. Diet is another important factor that is hard to decouple from chemical exposures since pesticide residues linger on much of the food we eat [16]. 

Given the complexity of the topic, it is essential to have predictive models that can provide accurate information on the factors that influence male infertility. Traditionally, the total motile sperm count (volume × concentration × motility) has been used as the most predictive factor in determining fertility compared to volume, concentration, and motility individually [17]. On the other hand, a recent study [18] reported that semen analysis (sperm quality) is a fundamental method for evaluating male fertility, including the analysis of sperm count, motility, morphology, and volume. Therefore, abnormalities in these parameters can indicate male infertility.

Since many factors have been investigated as potential risk factors related to male infertility [5,7,9,10,11,12,13,14,15,16,17], it is important to identify statistical models that can accurately predict how much each single risk factor impacts the onset of male infertility.

The introduction of artificial intelligence (AI) into the healthcare field promoted a significant change in the approach of medical practitioners to diagnosing, treating, and anticipating health ailments. Indeed, the use of AI, which is powered by machine learning algorithms, data, and computational capabilities, allows for the analysis of large datasets with impressive speed [18]. This transformative technology can predict illnesses, categorize patient information, and offer tailored treatment recommendations. Predictive analytics in medicine leverages AI to forecast patient outcomes, identify potential health risks, and optimize care pathways using algorithms such as the Random Forest (RF) Classifier [19,20]. Exploiting different types of information AI models can forecast the commencement and advancement of diseases, such as in the study of heart disease [21], cancer [22], and diabetes [23], in which AI models have already proven their efficiency in facilitating timely intervention and tailored healthcare strategies. 

Machine learning (ML), artificial neural networks (ANNs), and deep learning have emerged as feeble tools, stagnating our approach to comprehending and addressing male reproductive health. These tools have been employed to scrutinize extensive datasets, thereby assisting in the identification of pivotal factors that influence fertility outcomes. 

Currently, in the scientific literature, there are not many review documents that have specifically addressed the use of machine learning models for predicting male infertility. A recent review [24] addressed the topic in a non-systematic and narrative way. The authors [24] drew up examples of possible applications of ML algorithms in a hospital setting, providing indications on robots and mechanics in vogue for specialized analyses, such as Advancing Computer-Assisted Semen Analyses (CASA), in the investigation of preoperative images before surgical sperm retrieval, for intra-operative advancement to identify genetic material at a microscopic level and sperm identification and analysis. This review, although concise and well argued, has the limitation of being focused on assisted reproductive technologies without any specific reference to machine learning models. Another review [25] analyzed the role of ML models exclusively for sperm selection before fertilization practices without considering other factors. Two other reviews instead investigated azoospermia in patients with Klinefelter Syndrome [26] or, in an all-encompassing way [27], about both male and female fertility, raising important consideration about the need for data augmentation, feature extraction, explainability, and the need to revisit the meaning of an effective system for fertility analysis. 

For this reason, it seems important to provide precise indications on what types of algorithms and their relevant clinical performances are currently used in the scientific panorama, being able to discriminate specific models and data in which they have been used. Among the multiple ML models currently available, artificial neural networks (ANNs) are widely used in medical settings to improve the delivery of care at a reduced cost [28]. These models are particularly interesting since they are inspired by the neural organization of the human brain, which is modelled as an interconnection of nodes. In the context of male infertility, ANNs have played a crucial role in predicting sperm concentration [29]. This predictive ability is invaluable in assessing sperm quality, an essential determinant of male fertility. 

Even though we have access to several computational predictive models, challenges remain, such as understanding which of all these models is or has the potential to be the most accurate overall. 

By conducting a review on the accuracy of ML, with a focus on ANNs, in predicting male infertility, this research will provide valuable insights into this emerging field. Therefore, this review does not aim to claim a technical contribution, but it is intended to be a reference guide. Therefore, the aim of this review is to fill this gap and contribute to the existing literature by critically evaluating the accuracy of ML in predicting male infertility and gaining insights about which model might be more efficient. 

## 2. Materials and Methods

We used the PICO framework [30] for formulating the research questions reported in Table 1. The literature search was undertaken between August 2023 and October 2023 by two independent researchers (VS, DDB) using three databases (PubMed, Scopus, and ScienceDirect). Therefore, a search string was created as similar as possible for every database. The search strategy was conducted through the combination of Mesh, Tiab, and synonym terms for the general string of “Man, infertility, prediction”. In Appendix A, we report specifications about Mesh, Tiab, synonyms (Table A1), and the specific strings search (Table A2) adapted for each database. This review followed the Preferred Reporting Items for Systematic Reviews and Meta-Analyses (PRISMA) checklist [31].

All results from Pubmed, Scopus, and Science Direct searches were aggregated in an Excel sheet, arranging the titles in alphabetical order. Since the aim of this review was to make a comprehensive panoramic of ML models used to predict male infertility, we did not introduce any new approaches or databases to claim a technical contribution. 

### 2.1. Inclusion Criteria

After removing duplicates manually, the search results underwent title and abstract screening, applying criteria for inclusion and exclusion of search results as follows:-Inclusion: Studies published in a peer-reviewed journal; any year of publication; all study designs; human male population.-Exclusion: Any language other than English; grey literature, letter to the editor, or reviews; search results with content not directly relevant to the research question after; undergoing a title and abstract screening; studies with different target populations (female, animals); articles with ambiguity in the context of male infertility in humans and machine learning; paper with no direct access to the full text.

The reference lists of the included studies were screened for further relevant publications. For a paper in which animal and male models are mutually included, we considered only data relating to human male infertility. 

### 2.2. Quality of the Studies and Risk of Bias Analysis

Two independent researchers analyzed the quality (VS) and the Risk of Bias (RoB) (DDB) of the included studies. The selected studies were grouped following the study design, and then a suitable tool for quality appraisal had to be chosen according to the study design. Retrospective and prospective studies were both assessed with the JBI Checklist for Case Series, which consists of a rapid assessment based on the identification of ten specific items of information regarding the participants (from enrolment to the final stage of the experiment) and the procedure [32]. The remaining studies were assessed according to the EQUATOR guidelines [33]. Indeed, cohort, case–control, and cross-sectional studies were assessed through the STROBE guidelines for reporting observational studies. This tool is made of 22 items, assessing in detail all the sections of the paper and also including sub-items to further distinguish specific criteria of scoring according to the study design (cohort, case–control, cross-sectional) for the participants, the statistical methods, and the results sections. Studies using a multivariable prediction model were assessed through the Transparent Reporting of a multivariable prediction model for Individual Prognosis or Diagnosis (TRIPOD) checklist [34]. Similar to the STROBE, this tool is made up of 22 items including sub-items, thus collecting a score made up of 35 investigated domains.

For each study, we collected a score based on the screening performed using the suitable tool. This scoring was then normalized to the total, expressed as a percentage, and we called it quality grade. This normalization was applied to all the studies included in this review and allowed us to make a direct comparison of studies with different designs. Since these tools do not provide a cut-off score for further classification of the study quality, according to a previous study [35], we considered it as good quality an assessed adherence ranging from 60 to 80%.

The RoB analysis was performed through the administration of the prediction model risk of bias assessment tool (PROBAST) [36]. This instrument is suitable for reviews of studies about clinical prediction models. The PROBAST checklist assesses the risk that arises from the methods used and the consequential applicability of the prediction model.

## 3. Results

From a total of 254 results in the above-mentioned database searches, 39 duplicates were removed manually, and 215 results remained for title and abstract screening. Of them, 142 records had to be ruled out after applying the exclusion criteria on titles and abstracts, and the remaining 73 records were sought for retrieval. It was not possible to get full access to 1 of them, and, thus, the full-text screening of 72 studies was performed according to the inclusion and exclusion criteria. From the full-text reading, 29 papers were excluded for a final 43 eligible studies included in this review, as reported in the flowchart in Figure 1.

Since the studies are very different from each other, it was not possible to carry out a meta-analysis. Therefore, to answer the PICO questions formulated in the previous section, we organized and schematically summarized them in the upcoming tables. Indeed, the studies were divided based on the general topic they dealt with; therefore, the specific variables considered in each model included sperm retrieval (Table 2, four studies [37,38,39,40]); sperm quality, further divided into the investigation of sperm quality and morphology (Table 3a, seventeen studies [41,42,43,44,45,46,47,48,49,50,51,52,53,54,55,56,57]) and quality of sperm and environmental factors (Table 3b, four studies [58,59,60,61]); non-obstructive azoospermia (Table 4, three studies [62,63,64]); IVF outcome (Table 5, three studies [65,66,67]); environmental and medical factors (Table 6, twelve studies [68,69,70,71,72,73,74,75,76,77,78,79]) as a function of male infertility, according to a recent review addressing the broad causes and risk factors of male infertility [80]. For the sake of brevity, we decided to shorten the content in the tables as much as possible. Therefore, many acronyms normally used in AI and ML papers have been adopted. Indeed, to avoid repetition in the caption of the table, we decided to put a detailed glossary of these terms in the Abbreviations section.

We found that 86% of the included studies in this review used an ML algorithm to predict male infertility, while the remaining 14% were studies about the development of an ML algorithm [37,38,39] or for the purpose of predicting IVF outcome [20,42].

In these studies, about forty different AI and ML methods were used, most of the time in combination with other methods (mean = 3.1, IQR = 1–9). Two studies [37,43] reported the generic term ML, while in eight studies [42,43,45,46,47,51,59,61], algorithms available on the industrial market were used. Of the 40 models analyzed, most are models with 73% supervised learning, 20% are unsupervised, 3% are semi-supervised, and the remaining 5% are models that can be programmed in both modes, as in the case of ANNs.

Unsupervised ML models were always used in conjunction with those supervised to create classifications or compare ML algorithms’ performance. Found as the most used models, support vector machine (SVM, 11%), Random Forest (RF, 10%), linear regression (LR), artificial neural networks (ANNs), and multi-layer perceptron (MLP) were employed with a frequency of 7%. eXtreme Gradient Boosting (XGBoost), Least Absolute Shrinkage and Selection Operator (LASSO—GFLASSO), Convolution Neural Network (CNN), and K-Nearest Neighbor (KNN) had a frequency of 4%. The remaining 31 models were used in percentages ranging from 3 to 1%. Despite the variety of these studies, from the different data sources processed, and the methods used, it seems very clear that ML algorithms can process a huge amount of data, as in the studies [39,43,68,79] in which the sample size exceeded one thousand units, and multiple variables were combined for the prediction. However, this does not imply that a lower number of inputs corresponds to lower model efficiency [52,55,56].

Finally, since the aim of this review was to understand not only which models were used but also what their actual level of prediction accuracy was, we analyzed the results of the reported accuracy, considering as a primary outcome measure the tested accuracy of the model in the prediction of male infertility. In six studies [43,47,55,56,57,76], the performance of each model was not evaluated as accuracy, but an error estimation of the model prediction was made; in two studies [74,75], there was no information on the matter. In the remaining 35 studies, there was an average accuracy of 88% (IQR = 80–94%), with 26 studies declaring an accuracy greater than 80%. Considering the average of the accuracy values reported in the individual studies, we found that Boruta, DNN, and KNN had an accuracy of 99%, AdaBoost 97%, DMTL, and DTLA of 94%,

FSNN, LDFA, and QDFA were at 93%, U-NET and NNET architectures 92%, Bayesian models (NB—BKMR) 91%, and all other models reported an accuracy between 89 and 71%. Studies in which ANNs were used reported a mean accuracy level of 84% (IQR = 81–94%), showing a promising role for the adoption of this model for male infertility prediction.

### 3.1. Retrospective and Prospective Studies

In this review, 63% of the included papers were retrospective or prospective studies and, therefore, screened with the JBI critical appraisal quality tool. They reported, on average, a good quality (mean = 90%, IQR = 80–100%), while a more in-depth analysis performed with the RoB tool (PROBAST) revealed a lot of uncertainty or high concern (Figure 2).

Most of the studies reported an overall unclear applicability, mainly due to missing information about inclusion and exclusion criteria in the selection of data sources (60%). This is mainly due to the use of datasets when authors do not report specific information about the selection of records but merely information about data extracted for their analysis. Further, 40% of the included studies in this category also reported a high concern due to the applicability of the chosen analysis because no information is available about the distinction of predictors in the outcomes of the study. This does not allow us to infer the causality of results, which are, most of the time, summarized in terms of probability, association, and rates. Despite these considerations, the results are properly reported in most of the studies (93%), even if, most of the time, the results are mainly related to the ML algorithm, i.e., its validation or comparison with other models, and no specific outcomes are related with respect to the data sources. For this reason, the overall risk of bias is unclear or with high concern.

### 3.2. Observational Studies

Furthermore, 25% of the included papers were clinical research with an observational design and were assessed according to the STROBE guidelines. They reported, on average, a good quality (mean = 86%, IQR = 77–91%), which mainly reflects the RoB analysis. Only one study [66] reported high concern for both the overall Risk of Bias and applicability score. This study [66] combined ML algorithms with data mining in its method. Indeed, the authors used mixed data to predict the increment in the rate of implantation, also computing the classification of sperm quality. The results are also not properly reported in terms of prediction of the determined outcomes and, instead, as a function of the goodness of performance computed for each algorithm. As for the previous group of papers, these studies are also missing specific information about the enrollment and selection of participants, which may lead to an unclear RoB due to participant selection. On the other hand, the methods are well described, so 73% of these studies have a low concern in this regard (Figure 3).

### 3.3. Multivariable Prediction Model Studies

The remaining 12% of papers were studies on multivariable prediction models, assessed using the TRIPOD tool. They report an average adherence of 67%. In this small group of studies, no one reported high concern in any of the overall RoB and applicability scores. Since these studies properly set their aim as the diagnostic/prognostic prediction of male infertility, PROBAST assessment is probably more in line with these categories of studies than the others reported above (Figure 4).

## 4. Discussion

During the last few years, researchers have been exploring the use of machine learning (ML), artificial neural networks (ANN), and deep learning (DL) techniques to predict male infertility. This review covers a diverse array of studies and presents a comprehensive overview of the published literature in the field of male infertility prediction using machine learning. Advancements in the prediction of male infertility have been witnessed through the integration of machine learning (ML), artificial neural networks (ANN), and deep learning techniques. Noteworthy studies, such as the one proposed by Chen A. [81] based on region-based convolutional neural networks (R-CNNs), exemplify the potential of deep learning architectures in the assessment of sperm morphology.

Some studies [58,62,63,66] are actually comparative analyses, encompassing the utilization of diverse AI techniques, including ANNs, DT, and SVM, for the detection of male fertility. These methodologies demonstrated differing degrees of accuracy, emphasizing the significance of choosing the suitable algorithm for specific tasks. ANNs, in association with the LR, have exhibited promise in the prediction of biochemical parameters associated with male infertility [63]. The adaptability of ANNs permits one to consider intricate relationships within the data, thereby facilitating precise predictions. Furthermore, deep learning techniques have been investigated for the prediction of sperm fertility. Machine learning frameworks, such as the one proposed for the automatic prediction of human semen parameters [60], offer potential applications in assisted reproductive technologies. In addition, the application of machine learning-based spectrophotometry for quantifying sperm concentration presents a rapid and cost-effective technique [52].

Certain studies have the limitation of being vaguely defined [54,58,59,66,70,71], thereby posing challenges in assessing the robustness of predictive models. It is crucial to clearly delineate the limitations of the study for appropriate interpretation of the results.

This is the first review that attempted to collect all the studies addressing the use of machine learning to predict male infertility, including multiple approaches and different data sources to build the model. This meant that even if our intent was ambitious, it encompasses some limitations. Indeed, during the screening of the studies, we found incomplete or insufficient information about datasets or the adopted method, which impeded a punctual analysis of the accuracy of the ML models used in the studies included in the present review.

Machine learning (ML) models, despite their effectiveness, may lack interpretability, thereby making it difficult to comprehend the reasoning behind the predictions. It is imperative to ensure the transparency and interpretability of models to instill trust in their clinical applications. Indeed, in the included studies, we found huge differences in the input information used to build models. As we reported in Table 1, Table 2, Table 3, Table 4, Table 5 and Table 6, the data sources used for building each model were very different, including demographic data, RNA, quality of the embryo, sperm retrieval, hormonal data, and others. This variety, in addition to the fact that, in many studies, multiple variables were used as a combination matrix, limited possible direct comparisons among the models, also leading to different results. This also means that it is not possible to make a precise statement about what variables are strictly correlated to the pathogenesis of male infertility, since each model made a unique combination of the set variables to be included for prediction. Further reviews may face this problem, formulating a specific question that could be addressed with a meta-analysis.

Another limitation of our study is that a mathematical discussion of the ML algorithms used in the reviewed studies is missing. Although an interesting point of view, this aspect was excluded because it was out of the scope of this review and because it would need considerable work, most suitable for a book or an article collection.

Despite the focus of this review being precisely to analyze ANN models, we found that only a few studies used this model for this purpose. However, even if they are a small number, it seems that these studies have a lower RoB, like those of Ma et al. [40] and Girela et al. [60], who obtained low concern regarding the overall applicability of the methods.

Finally, considering that the ANN was associated with a high level of accuracy in the prediction of male infertility, we can hypothesize that it has potential, so its use may increase in the future.

## 5. Conclusions

Models based on artificial neural networks (ANNs) have demonstrated potential in predicting male fertility, with a few achieving remarkable accuracy. One ANN model achieved a maximum accuracy of 95% in predicting male fertility utilizing explainable AI [77]. Deep learning techniques, a subset of ML, have been employed to evaluate sperm motility and morphology [44,45,68]. Unlike traditional methods, CNNs excel at processing visual data, making them particularly well suited for tasks involving embryo morphology and have also demonstrated effectiveness in assessing spermatozoid motility [44,45,47,53]. Certain investigations might employ a combination of ML, ANN, and DL techniques for a comprehensive analysis, harnessing the strengths of each.

Choosing one model based only on the highest accuracy reported might be an inappropriate decision since it is crucial to consider the nature of the data, the complexity of the prediction task, and the interpretability of the results when selecting the most suitable model. We conclude that the use of ML models for the identification of risk factors and prediction of male infertility has the potential to assist personalized medicine in achieving its goal of creating optimally tailored diagnostic, preventive, and therapeutic measures. This review had the ambitious intent to collect all the studies in which ML models were used to predict male infertility to support those who are approaching the use of machine learning in the medical field for the first time in choosing the most suitable model, based on the accuracy levels already established by previous research. Future studies on male infertility prediction may consider other computational methods, such as the exploitation of the graph contrastive clustering techniques [82] that are based on the use of deep neural networks for representation learning before clustering. This approach, for example, has the advantage of bringing similar or positive sample pairs closer and pushing dissimilar or negative sample pairs further away, thus going beyond the clustering methods of unsupervised machine learning algorithms.

Finally, considering that claiming a technical contribution was out of the scope of this review, we think that further research should address this point by introducing new approaches or databases.

## Figures and Tables

**Figure 1 healthcare-12-00781-f001:**
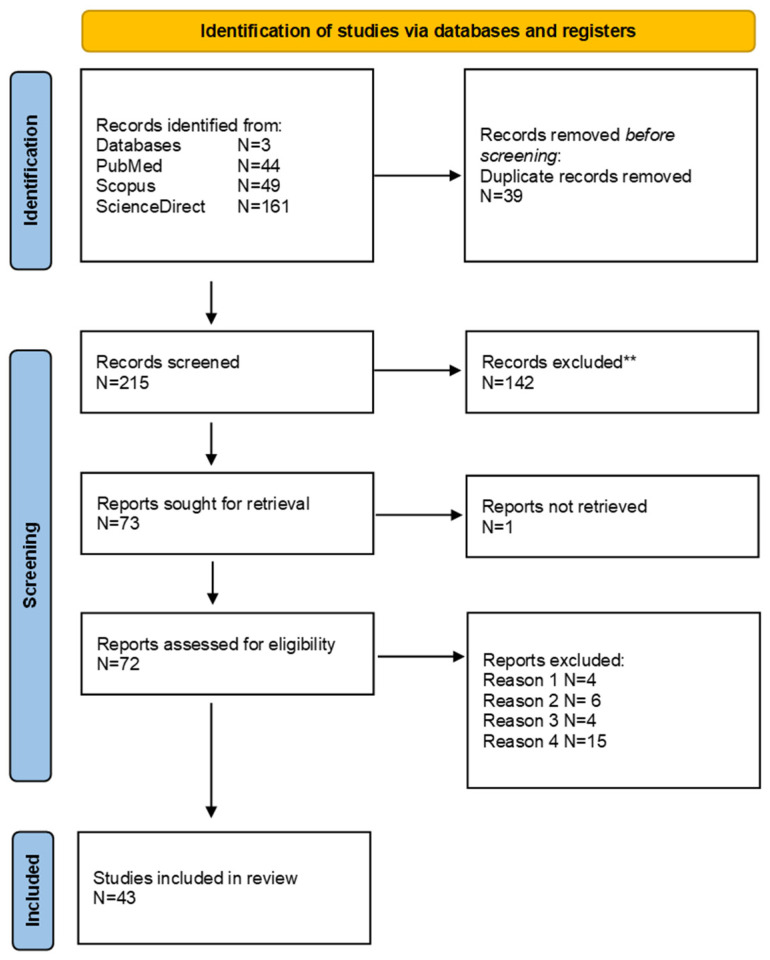
PRISMA flowchart diagram of the review. ** Reasons for the 142 records exclusion were: (1) used animals or females as participants; (2) did not use ML; (3) the format of the paper was review, letter to the editor, or others; (4) the aim of the study was out of scope concerning the prediction of male infertility.

**Figure 2 healthcare-12-00781-f002:**
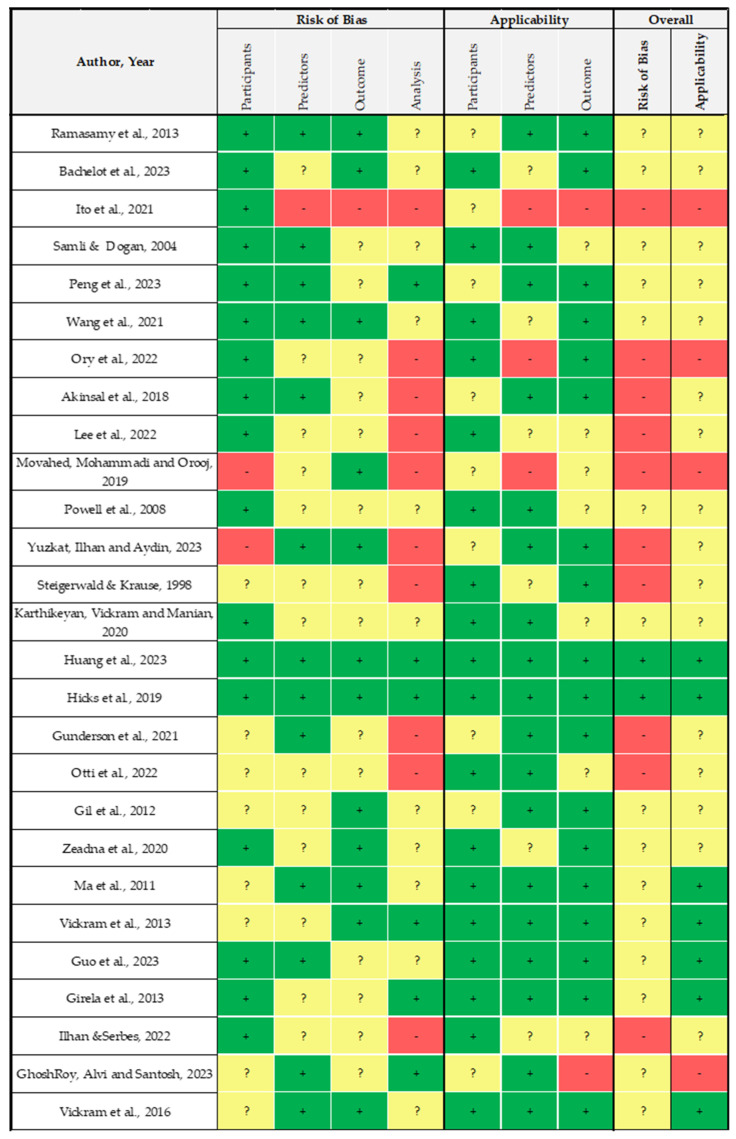
Risk of Bias (RoB) analysis of retrospective and prospective studies performed using PROBAST [37,38,39,40,41,42,43,44,47,48,54,55,56,57,58,59,60,61,63,64,71,72,74,75,76,77,79].

**Figure 3 healthcare-12-00781-f003:**
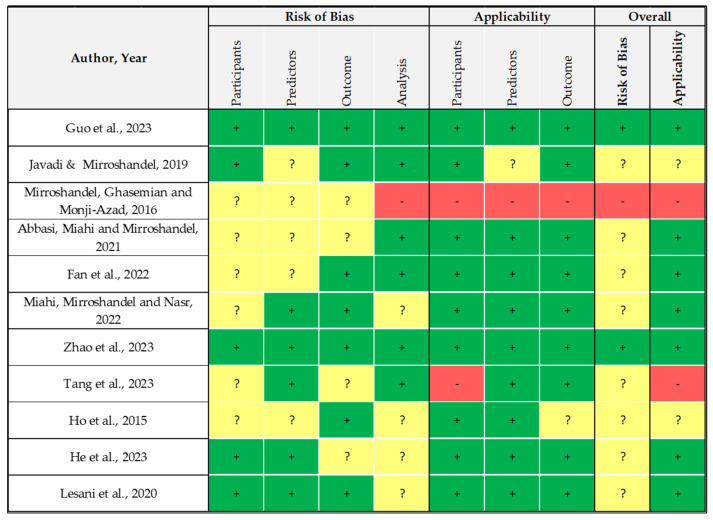
Risk of Bias (RoB) analysis of observational studies performed using PROBAST [45,46,49,52,53,65,66,68,69,70,78].

**Figure 4 healthcare-12-00781-f004:**
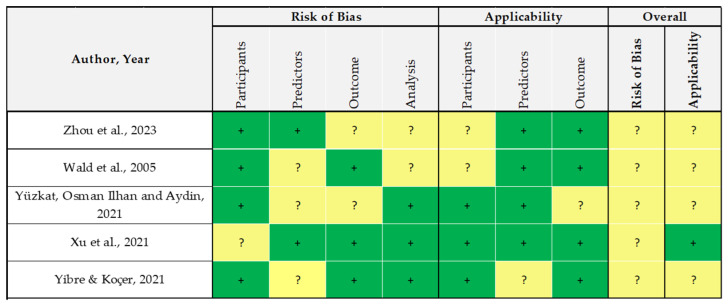
Risk of Bias (RoB) analysis of multivariate prognostic and diagnostic studies performed using PROBAST [50,51,62,67,73].

**Table 1 healthcare-12-00781-t001:** PICO framework.

Acronym	Definition	Motivation	Research Question
P	Population	Gain insights about the predictive tools using ML for Male infertility	Are there any models that predict male infertility?
I	Intervention	Analyze state of the art of the different ML and ANN algorithms used to predict male infertility	Which computational models have been used to predict Male infertility?
C	Comparison	Compare the different algorithms and the features/indicators	Which of these computational models require fewer features in the algorithm?
O	Outcome	Understand the prediction accuracy of the algorithms	What is the accuracy of the ML in comparison to other predictive models?

**Table 2 healthcare-12-00781-t002:** In this table, studies in which ML models were employed to investigate the relationship between male infertility and the potential sperm retrieval as a risk factor for male infertility are grouped. Quality grade is the normalized scoring obtained through the administered chosen tool according to the criteria exposed in Section 2.2. Accuracy of the model refers to the tested accuracy of the model in the prediction of male infertility. Abbreviation: BMI stands for body mass index; LH is luteinizing hormone; FSH is follicle-stimulating hormone; ICSI is intracytoplasmic sperm injection; TESE is testicular sperm extraction; T is testosterone; AUC is the area under the curve; NOA is non-obstructive azoospermia.

Authors	Quality Grade	Algorithms Used	Data Source	Outcome	Accuracy
Bachelot et al., 2023 [37]	80%	ML, SVM, RF, GBT, XGB, LR, DL, KNN	Age, BMI, tobacco consumption, FSH and LH assessment, T, inhibin B, prolactin, karyotype and search for Y-chromosome microdeletion, urogenital history (cryptorchidism, infection, trauma, gonadotoxic therapy, urogenital surgery, and varicoceles).	The presence/absence of spermatozoa after examination of the surgical specimens. A positive outcome was defined as obtaining enough spermatozoa for the ICSI procedure.	The models achieved an accuracy greater than the 60%, with the best performance of RF (84.6%), GBT (76.9%), and XGB (80.8%).
Zeadna et al., 2020 [38]	80%	GBT, MvLRM	Baseline hormonal profile (before TESE) of serum FSH, LH, and T	The cutoff value for successful sperm retrieval was the presence of at least one viable of mature sperm in the testicular tissue.	AUC = 0.807 for predicting the presence of spermatozoa in patients with NOA.
Ramasamy et al., 2013 [39]	100%	ANN, LR	Clinical and laboratory data of sperm extraction	Development of an ANN and nomogram to predict sperm retrieval with microdissection testicular sperm extraction.	Nomogram accuracy: 59.6% ANN accuracy: 59.4%
Ma et al., 2011 [40]	90%	Three ANNs with feed forward-back propagation architecture were used	Leptin and FSH level	Leptin resulted in a good assistant marker for NOA diagnosis. ANNs improved the prediction accuracy of sperm retrieval.	ANN1 performance resulted in the best in the prediction of sperm recovery in NOA patients (AUC = 0.83).

**Table 3 healthcare-12-00781-t003:** (**a**) In this table, studies in which ML models were employed for investigating sperm quality e/o morphology as a function of male infertility are grouped. Quality grade is the normalized scoring obtained through the administered chosen tool according to the criteria exposed in Section 2.2. Accuracy of the model refers to the tested accuracy of the model in the prediction of male infertility. Abbreviation: CASA is Computer Assisted Sperm Analysis, DFI is DNA fragmentation index; GC-MS is gas chromatography-mass spectrometry; AAA is Artificial Algae Algorithm; SMA is sperm morphology analysis; MAE is mean absolute error. (**b**) In this table, studies dealing with the sperm quality analysis are grouped, thus exploring the environmental factors that may be associated with predisposition with male infertility. Quality grade is the normalized scoring obtained through the administered chosen tool according to the criteria exposed in Section 2.2. Accuracy of the model refers to the tested accuracy of the model in the prediction of male infertility.

(a)					
Authors	Quality Grade	Algorithms Used	Data Source	Outcome	Accuracy
Guo et al., 2023 [41]	91%	LR, RF, SVM	Metabolomics and proteomics using CASA and DFI	Proteins related to energy metabolism and oxidative stress were found to be differential biomarkers.	Estimated accuracy for each algorithm was 87% each
Yuzkat, Ilhan and Aydin, 2023 [42]	100%	YOLOv5 Deep Learning Based Object	Dataset including 12 sperm specimen videos	YOLOv5 achieved the best results in the first and second scenarios.	95% for almost all the videos.
Huang et al., 2023 [43]	90%	SMOreg, ML, RF, SGB, LASSO, Ridge, XGBoost	85 videos of human semen samples and related participants’ data.	ML-based analysis predicted sperm motility. The addition of participants’ data did not improve the algorithm’s performance.	The two-stream NNs were not significantly better than the baseline one.
Lee et al., 2022 [44]	70%	CNN based on U-Net architecture	3-channel input of size 256 × 256 pixels.	The algorithm detects and locates individual sperm cells.	Precision 84.0%, Sensitivity 72.7%
Fan et al., 2022 [45]	68%	DeepResolution2 (DNN), U-Net4S, k-CNN, U-Net4R models	GC-MS data files—Untargeted dataset	DeepResolution2 outperformed other methods in peak identification and quantification	AUC = 0.99
Miami, Mirroshandel, and Nasr, 2022 [46]	91%	Genetic Neural Architecture Search (GeNAS)—GeNAS Weighting Factor (GeNAS-WF)	MHSMA dataset—images of human sperm cells	Crossover operation helped GeNAS change the length of chromosomes.	91.7% in vacuole detection, and 77.7% in acrosome detection.
Otti et al., 2022 [47]	90%	Sparse optical flow with Lucas-Kanade algorithm, Crocker-Grier algorithm (MLP, RNN, CNN)	A dataset composed of semen analysis and related participants’ data	Improved prediction of sperm motility compared to the previous state of the art published data.	The MAE was reduced from 8.83 to 7.31.
Ilhan & Serbes, 2022 [48]	80%	Two-stage fine-tuned DNN: soft voting decision level ensemble learning scheme	Sperm Morphology Image Data Set (SMIDS), Human Sperm Head Morphology Set (HuSHeM) and SCIAN-Morpho.	The two-stage fine-tuning approach improves accuracy. The fusion of deep-nets results in higher precision scores.	90.9% for SMIDS, 88.9% for HuSHeM, and 72.1% for SCIAN-Morpho. The DNN increased the accuracy up to 92.1.
Abbasi, Miahi, and Mirroshandel, 2021 [49]	100%	DMTL, DTLA	Dataset of non-stained grayscale sperm images.	The algorithm automated sperm abnormality detection with improved accuracy in the identification of the head, acrosome, and vacuole.	Accuracy for vacuole labels reached the 93.75%.
Yüzkat, Osman Ilhan and Aydin, 2021 [50]	68%	CNN models Decision-level fusion techniques	Dataset of normal and abnormal sperm morphology images	The soft voting-based fusion approach achieved high classification accuracies for the three different data sets.	The accuracy of all tested models was greater than 94% for prospective azoospermic patients.
Yibre & Koçer, 2021 [51]	68%	AAA with learning-based fitness evaluation method, MLP, NB, SVM, KNN, RF	Dataset for prediction of semen quality—UCI public data source.	The outcome information from the automated medical diagnosis system is directly related to human health.	AUC = 0.975 for the classification of sperm quality.
Lesani et al., 2020 [52]	77%	ANN, FSNN	Full absorption spectrum data comprised 711 data points per sample.	The ML-based spectrophotometry approach accurately quantifies sperm concentration.	Over 93% accuracy in prediction.
Javadi & Mirroshandel, 2019 [53]	95%	Deep CNN, PCA, KNN	MHSMA dataset. Non-stained and low-resolution images.	The algorithm resulted seven times faster than SMA.	84.7% for acromosome, 83.9% for head, 94.6% for vacuole.
Movahed, Mohammadi and Orooj, 2019 [54]	100%	K-means clustering (CNN) and SVM classifier	Image data	The model outperformed previous works for head, acrosome, and nucleus segmentation.	Dice similarity of 0.90 for the head segment, 0.77 for the axial filament.
Vickram et al., 2013 [55]	100%	BPNN, mean squared error calculated for error propagation	Seminal fluid	Good correlation between estimated and predicted values (r = 0.9). Potential for Zinc prediction in human semen.	The MAE for the BPNN model was 0.025, −0.080, 0.166, and −0.057 for protein, fructose, glucosidase, and zinc, respectively.
Steigerwald & Krause, 1998 [56]	80%	ANN	CASA system with automatic determination of midpieces and sperm tails	Reproducible results for sperm morphology estimation	No significant difference in the % of normal forms compared to direct microscopical inspection.
Vickram et al., 2016 [57]	90%	BPNN model and RBFN	Semen samples from human participants	The BPNN model had an acceptable absolute error for predicting biochemical markers. The RBFN model had higher error compared to the BPNN model	Mean absolute error for BPNN model: 0.025, 0.080, 0.166, 0.057. RBFN model had higher error compared to BPNN model
(**b**)					
**Authors**	**Quality Grade**	**Algorithms Used**	**Data Source**	**Outcome**	**Accuracy**
GhoshRoy, Alvi, and Santosh, 2023 [58]	100%	SVM, RF, DT, LR, naive Bayes, Sdaboost, MLP	UCI datasets covering 9 inputs including environmental and lifestyle factors.	DT and RT models performed well, while SVM and naive Bayes provided poor prediction outcomes.	All the seven tested models achieved an accuracy higher than the 80% with the best performance of the RF classifier (96.7% of prediction).
Ito et al., 2021 [59]	80%	Google Cloud, AutoML Vision	Images of testicular tissues stained with hematoxylin and eosin.	Improved precision for Johnsen scores of 4–5 and 6–7 to 95 and 97.	At 400× magnification: 82.6% average precision of the algorithm with expansion images: 99.5%
Girela et al., 2013 [60]	80%	ANN, MLP	Sociodemographic, demographic, environmental, and health-related factors.	ANN predicted semen parameters with a high evel of accuracy in the prediction of sperm concentration and motility.	MLP showed a high accuracy in prediction of sperm concentration (93.3%) and motility (89.3%).
Gil et al., 2012 [61]	90%	C4.5 algorithm used for decision tree, MP, and SVM for prediction.	The data included information on environmental and lifestyle factors.	MLP and SVM showed the highest accuracy in prediction. Decision trees provided a visual and illustrative approach.	MLP and SVM achieved the highest accuracy (69%), being SVM the one with the higher Sensitivity (73.9%) whereas MLP obtained superior Specificity values (25%).

**Table 4 healthcare-12-00781-t004:** In this table, studies in which ML models were used for the diagnosis or investigation of non-obstructive azoospermia (NOA) are grouped. Quality grade is the normalized scoring obtained through the administered chosen tool according to the criteria exposed in Section 2.2. Accuracy of the model refers to the tested accuracy of the model in the prediction of male infertility.

Authors	Quality Grade	Algorithms Used	Data Source	Outcome	Accuracy
Zhou et al., 2023 [62]	74%	LASSO, Boruta, SVM-RFE, Random Forest	Transcriptome sequencing data of testicular cells. Immunohistochemical staining data for protein expression levels	An RF model based on the transcription factors ETV2, TBX2, and ZNF689 was successfully developed to diagnose NOA.	RF model achieved an AUC of 1000 and an F-measure of 1000.
Peng et al., 2023 [63]	90%	ANN, LASSO, SVM-RFE, LR, RF	RNA-binding protein-related genes. Testicular samples, clinical samples. scENA-seq data	An ANN diagnosis model based on RNA-binding proteins DDX20 and NCBP2 was developed. The ANN model exhibited reliable predictive performance in multiple cohorts	Training cohort (GSE9210) scored 74.1% of accuracy, GSE45885 the 90.3%, GSE45887 the 85.0%, while local cohort only the 59.1%
Samli & Dogan, 2004 [64]	100%	ANN, Logistic Regression	Patient age, duration of infertility, serum hormone levels, and testicular volumes	The NN correctly predicted the outcome in 59 of the 73 test set patients (80.8%)	The accuracy of the ANN model is 80%. The accuracy of the LR model is 66%.

**Table 5 healthcare-12-00781-t005:** In this table, studies in which ML models were used for the prediction of in vitro fertilization (IVF) outcomes are grouped. Quality grade is the normalized scoring obtained through the administered chosen tool according to the criteria exposed in Section 2.2. Accuracy of the model refers to the tested accuracy of the model in the prediction of male infertility.

Authors	Quality Grade	Algorithms Used	Data Source	Outcome	Accuracy
He et al., 2023 [65]	91%	WGCNA	Three azoospermia RNA chip datasets (GSE145467, GSE45885, and GSE9210), one COVID-19 RNA chip dataset (GSE157103), and one cryptozoospermia single-cell RNA-sequencing dataset (GSE153947) were downloaded from the NCBI GEO database	Screening of two different molecular subtypes revealed that azoospermia-related genes were associated with clinicopathological characteristics of age, hospital-free-days, ventilator-free-days, Charlson score, and d-dimer of patients with COVID-19	The accuracy of successful IVF was 0.72
Mirroshandel, Ghasemian and Monji-Azad, 2016 [66]	55%	Data mining, NB, SVM, MLP, IBK, Kstar, RC, J48, RF	Quality of zygote, embryo, and implantation outcome of injected sperms	Kstar model achieved the 95.1% in implantation outcome prediction	The RC model achieved the 83.8% of accuracy. The Kstar model the 95.9%.
Wald et al., 2005 [67]	68%	L & QDFA, LR, NNET	Maternal age, type of sperm retrieval, type of spermatozoa used (cryopreserved or “fresh”), and type of male factor infertility	The 4-hidden node NN model demonstrated high accuracy in predicting IVF/ICSI outcomes	The NN predicted intrauterine pregnancy with high accuracy (AUC = 0.923).

**Table 6 healthcare-12-00781-t006:** In this table, studies in which ML models were used for the prediction of male infertility as a function of environmental and medical factors are grouped. Quality grade is the normalized scoring obtained through the administered chosen tool according to the criteria exposed in Section 2.2. Accuracy of the model refers to the tested accuracy of the model in the prediction of male infertility. Abbreviation: MAE is mean absolute error, TIF is Tamura Image Features, and AUC is the area under the curve.

Authors	Quality Grade	Algorithms Used	Data Source	Outcome	Accuracy
Guo et al., 2023 [68]	90%	DLNM, BKMR	PREBIC cohort of semen samples from 3940 males.	Single- and two-pollutant models showed SO_2_, O_3_, PMs, and NO_2_ were negatively associated with progressive motility, total motility, and sperm morphology.	AUC = 0.889
Zhao et al., 2023 [69]	77%	SVM, XGB, GLM, and RF	Two microarray datasets (GSE4797 and GSE45885) related to male infertility (MI) patients with spermatogenic dysfunction.	Cuproptosis-related genes were found both in healthy and men with spermatogenic dysfunction.	XGB model based on 5-gene showed superior performance on the external validation dataset GSE45885 (AUC = 0.812)
Tang et al., 2023 [70]	86%	WGCNA RF, SVM, GLM, XGB	NOA microarray datasets (GSE45885, GSE108886, and GSE145467)	The model based on IL20RB, C9orf117, HILS1, PAOX, and DZIP1 biomarkers had the highest AUC value, of up to 0.982, compared to other single biomarker models.	XGB algorithm that had the maximum AUC value (AUC = 0.946)
Ory et al., 2022 [71]	80%	LR, RF, SVM	Pre and post-operative clinical and hormonal data following treatment	A total of 45.6% of men experienced an upgrade in sperm concentration following surgery, 48.1% did not change, and 6.3% downgraded	The RT-supervised machine learning model had good accuracy in the prediction of outcome (AUC = 0.72).
Gunderson et al., 2021 [72]	100%	RF, SGB, LASSO, Ridge, XGBoost	Annual health screening data	Ridge regression showed the best performance for SMAPE and RAE metrics	ML model predicted successful conventional IVF with a mean accuracy of 0.72.
Xu et al., 2021 [73]	58%	CNN, DPPCG	Human protein-coding genes from the NCBI database and human proteins from the UniProt database	DPPCG harnessed the utility of heterogeneous biomedical big data in the effective indirect prediction of 794 causal genes of male infertility and associated pathological processes.	The accuracy of the deep CNN models was 0.70, with an average precision of 0.74, and an average recall rate of 0.56.
Wang et al., 2021 [74]	100%	GFLASSO	138 environmental/ behavioral/ psychological variables and 32 male reproductive biomarkers in 796 young Chinese men.	Thirty-one of the thirty-two reproductive biomarkers had positive correlations with the predictive values, with an average correlation coefficient of 0.26, ranging from 0.10 to 0.40.	Not reported.
Karthikeyan, Vickram and Manian, 2020 [75]	80%	Data Mining	Semen samples from three different categories fertile (N = 20), Infertile (N = 20), and unilateral varicocele (N = 15) men.	There were 6 highly significant results: rs14988405 (R4W), rs201470131 (A52P), rs570385517 (R90C), rs17104534 (G240R), rs148319106 (V318M) and rs200608161 (V352L).	Not reported.
Hicks et al., 2019 [76]	100%	AdaBoost, GP, KNN, MLP-SKLearn, SVM, CatBoost and MLP-TensorFlow	Sequences of frames from video recordings of human semen under a microscope	Multimodal analysis methods combining video data with participant data did not improve the prediction of sperm motility compared to using only the video data.	RF was the best for participant data only (MAE = 11.368), for TIF only SMOreg (MAE = 10.800), TIF and participant ata RF scored a MAE = 11.617.
Akinsal et al., 2018 [77]	90%	MLP, ANN	Testicular volume, follicle-stimulating hormone, luteinizing hormone, total testosterone, and ejaculate volume of the patients.	Total testicular volume with LH had the highest power to find out which participant requires sex chromosome evaluation.	LR analyses and ANN predicted the presence-absence of chromosomal abnormalities with more than 95% accuracy.
Ho et al., 2015 [78]	77%	LDA	Detection and interpretation of pathogenic copy number variants of gonadal function.	Protein–protein interactions were the most informative for gene prediction, followed by gene expression and epigenetic marks	The AI scored an AUC of 0.711 in the classification of candidate genes.
Powell et al., 2008 [79]	100%	NN, LR, discriminant FA	Testis volume, sperm density, motility, and the presence of endocrinopathy.	LR and a neural network performed the best with receiver operating characteristic areas under the curve of 0.93 and 0.95	LR and the NN performed the best with AUC=0.93 and 0.96 respectively.

## Data Availability

The data that support the findings of this study are available from the corresponding author upon reasonable request.

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
