# Peer review of "Predicting Male Infertility Using Artificial Neural Networks: A Review of the Literature"

_healthcare, 2024, doi:10.3390/healthcare12070781_

Round 1

Reviewer 1 Report

Comments and Suggestions for Authors

Dear author. Thank you for an excellent manuscript. The area of male infertility is by all means underrated and emphasis on the topic is well appreciated. The field of AI has been studied extensively in the field of reproductive medicine and such a review is truly not available at the moment. I would like to see the paper published. 

However, there are few minor issues that have to be settled.

1. The language is generally good but there are several improvements required. 

2. Please extend the limitation section with some explanation how several studies build models based on very different input information. Even though your focus is on ML models, studies differ significantly on observed data and it has to be very clear.

Comments on the Quality of English Language

Even though I am not a native speaker, I find several minor issues with the language that should be improved. 

Author Response

We would like to thank the Editor and the Reviewers for their positive general judgments about our manuscript and for their qualified comments that helped us to improve our work in its revised version. In the following our point-by-point responses with the changed parts of the manuscript are reported between apices and in italic style and highlighted in the revised version of the manuscript by word change tracking.

REVIEWER #1
Dear author. Thank you for an excellent manuscript. The area of male infertility is by all means underrated and emphasis on the topic is well appreciated. The field of AI has been studied extensively in the field of reproductive medicine and such a review is truly not available at the moment. I would like to see the paper published. 

Authors: Thanks for understanding the importance of this topic and for the very positive comment about our work

However, there are few minor issues that have to be settled.

  1. The language is generally good but there are several improvements required. 

Authors: As suggested by all reviewers, we made an English edit of the manuscript

  1. Please extend the limitation section with some explanation how several studies build models based on very different input information. Even though your focus is on ML models, studies differ significantly on observed data and it has to be very clear.

Authors: We thank the reviewer for this suggestion. We had already presented this information in tables (from 1 to 4) and now we added the following sentence to stress the limitation that this fact necessarily implies.

Indeed, in the included studies we found huge differences in input information used to build models. As we reported in tables ranging from 1-4, the data sources used for building each model were very different, including demographical data, RNA, quality of the embryo, sperm retrieval, hormonal data, and others. This variety, in addition to the fact that in many studies multiple variables were used as a combination matrix, contributed to limiting possible direct comparison among models, also leading on different results.”

Reviewer 2 Report

Comments and Suggestions for Authors

The authors explore a timely development in male infertility diagnosis that demands exploration since the use of artificial intelligence in medicine is an uprising issue in terms of accuracy and conditions of its incorporation in routine practice. 

The authors have presented a robust review design by incorporating the PICO and ROB framework and their presentation and methodological approach valids the quality of the manuscript. 

The only point I would like to comment on is that the authors should review the manuscript for some minor deviations in reporting style as for example sexual encounter should be reported as sexual intercourse instead and in the abstract first line there is "not" systematic review should be there is no systematic review. Please watch out for such minor mistakes throughout the text. 

Comments on the Quality of English Language

Some minor corrections required as denoted in the comments to authors. 

Author Response

We would like to thank the Editor and the Reviewers for their positive general judgments about our manuscript and for their qualified comments that helped us to improve our work in its revised version. In the following our point-by-point responses with the changed parts of the manuscript are reported between apices and in italic style and highlighted in the revised version of the manuscript by word change tracking.

REVIEWER #2
The authors explore a timely development in male infertility diagnosis that demands exploration since the use of artificial intelligence in medicine is an uprising issue in terms of accuracy and conditions of its incorporation in routine practice. 

Authors: Thanks for having get the importance of this topic and for the general positive comment about our work

The authors have presented a robust review design by incorporating the PICO and ROB framework and their presentation and methodological approach valids the quality of the manuscript. 

Authors: We thank the reviewer for his/her positive judgment on the adopted methodological approach in this paper.

The only point I would like to comment on is that the authors should review the manuscript for some minor deviations in reporting style as for example sexual encounter should be reported as sexual intercourse instead and in the abstract first line there is "not" systematic review should be there is no systematic review. Please watch out for such minor mistakes throughout the text. 

Authors: As suggested by all reviewers, we made an English edit of the manuscript. Then, for the specific comments, we have modified “sexual encounters” in “sexual intercourses”, and “not” was changed to “no”

Reviewer 3 Report

Comments and Suggestions for Authors

The study aims to comprehensively investigate ML in predicting male infertility. A comprehensive literature search was conducted in PubMed, Scopus and Science Direct between July 15th and October 23rd, 2023, conducted in accordance with the Preferred Reporting Items for Systematic Reviews and Meta-Analyses (PRISMA) guidelines. In addition, the discussion part should be more detailed and credible. The followed comments need to be clarified.

1. The deficiencies in the previous reviews are not sufficiently described, which makes the motivation of the manuscript unclear.

2. In lines 163 to 168, the authors classify the existing literature. What is the basis for this classification way?

3. Please explain some information in the tables, such as "Quality Grade".

4. This manuscript discusses the use of artificial neural networks in the field of predicting male infertility, but only a small percentage of the selected papers seem to be ANN-based.

5. The authors discuss common computational methods such as ANNs and deep learning in the manuscript. In future work, the authors may consider the use of other computational methods in this field, such as graph representation learning (10.1109/JBHI. 2024.3357979).

Comments on the Quality of English Language

 Grammar and words in this manuscript need to be improved, such as "th6" in line 173.

Author Response

We would like to thank the Editor and the Reviewers for their positive general judgments about our manuscript and for their qualified comments that helped us to improve our work in its revised version. In the following our point-by-point responses with the changed parts of the manuscript are reported between apices and in italic style and highlighted in the revised version of the manuscript by word change tracking.

REVIEWER #3

The study aims to comprehensively investigate ML in predicting male infertility. A comprehensive literature search was conducted in PubMed, Scopus and Science Direct between July 15th and October 23rd, 2023, conducted under the Preferred Reporting Items for Systematic Reviews and Meta-Analyses (PRISMA) guidelines. In addition, the discussion part should be more detailed and credible. The following comments need to be clarified.

Authors: Thanks for understanding the importance of this topic and for the general positive comment about our work

  1. The deficiencies in the previous reviews are not sufficiently described, which makes the motivation of the manuscript unclear.

Authors: we thank the reviewer for this valid comment. We tried to outline the deficiencies of previous reviews on this topic also stressing the motivation of our current work, by adding the following paragraph in the main text of the manuscript.

Currently, in the scientific literature, there are not many review documents that have specifically addressed the use of machine learning models for predicting male infertility. A recent review [24] addressed the topic in a non-systematic and narrative way. The authors have drawn up examples of possible applications of ML algorithms in the hospital setting, providing indications on robots and mechanics in vogue for specialized analyses such as Advancing Computer-Assisted Semen Analyses (CASA), in the investigation of preoperative images before surgical sperm retrieval, for intra-operative advancement to identify genetic material at a microscopic level and sperm identification and analysis. This review, although concise and well-argued, has the limitation of being focused on assisted reproductive technologies without any specific reference to machine learning models. Another review [25] analysed the role of ML models exclusively for sperm selection before fertilization practices without considering other factors. Two other reviews have instead investigated azoospermia in patients with Klinefelter Syndrome [26] or in an all-encompassing way [27] about both male and female fertility, raising important consideration about the need for data augmentation, feature extraction, explainability, and the need to revisit the meaning of an effective system for fertility analysis.

For this reason, it seems important to provide precise indications on what types of algorithms and their relevant clinical performances are currently used in the scientific panorama, being able to discriminate specific models and data on which they have been used.

  1. In lines 163 to 168, the authors classify the existing literature. What is the basis for this classification way?

Authors: We thank the reviewer for this comment. More than a classification it would be a schematic reporting of the included studies to organize the content according to the specific topic they dealt with, thus making an organized presentation of the results as reported in the following sentences. 

Therefore, to answer the PICO questions formulated in the previous section, we have organized and schematically summarized them in the upcoming tables. Indeed, the studies were divided based on the general topic they dealt with, i.e.: Sperm Retrieval (table 2, four studies [37-40]), Sperm Quality further divided into the investigation of Sperm Quality & Morphology (table 3.1, seventeen studies [41-57]) and Quality of Sperm & Environmental Factors (table 3.2, four studies [58-61]), Non-Obstructive Azoospermia (table 4, three studies [62-64]), IVF outcome (table 5, three studies [65-67]), Environmental & Medical factors (table 6, twelve studies [68-79])

This organisation also reflects a previous review of factors predicting male infertility which now we reported in the main text to motivate our choices as follows: “according to a recent review addressing the broad causes and risk factors of male infertility [80]”.

  1. Please explain some information in the tables, such as "Quality Grade".

Authors: Thanks for this suggestion, in the revised version of our manuscript we have added a description in each table to address the comment of the reviewer, as follows: “Quality grade is the normalized scoring obtained through the administered chosen tool according to the criteria exposed in paragraph 2.2.” The explanation is quite long to be extensively reported into the legend, but we have clarified that the reader could have details and information in paragraph 2.2 where is written:

“(…) Retrospective and prospective studies were both assessed with the JBI Checklist for Case Series which consists of a rapid assessment based on the identification of ten specific information regarding the participants (from enrolment to final stage of the experiment) and the procedure [32]. The remaining studies were assessed according to the EQUATOR guidelines [33]. Indeed, cohort, case-control, and cross-sectional studies were assessed through the STROBE guidelines for reporting observational studies. This tool is made of 22 items assessing in detail all the sections of the paper also including sub-items to further distinguish specific criteria of scoring according to the study design (cohort, case-control, cross-sectional) for the participants, the statistical methods, and the results sections. Studies using a multivariable prediction model were assessed through the Transparent Reporting of a multivariable prediction model for Individual Prognosis or Diagnosis (TRIPOD) checklist [34]. Similar to the STROBE, this tool is made up of 22 items including sub-items, thus collecting a scoring made up of 35 investigated domains.. (…)”

  1. This manuscript discusses the use of artificial neural networks in the field of predicting male infertility, but only a small percentage of the selected papers seem to be ANN-based.

Authors: the reviewer has properly raised his concern in this regard. Our first attempt was to identify studies in which ANN-based models were used to predict male infertility. Unfortunately, only 7 studies used these ANN models and we have now reported on this aspect with the following sentences:

Despite the focus of this review was precisely to analyse the use of ANN models in male infertility prediction, we found that only 7 studies have used this model for this purpose. However, even if they are a small number, it seems that these studies have a lower RoB, as those of Ma et al. [40] and of Girela et al. [60] which obtained a low concern regarding the overall applicability of the methods”.

Finally, considering that ANN was associated with a high level of accuracy in the prediction of male infertility, we can hypothesize that it has the potential so that its use may increase in the future.

Models based on artificial neural networks (ANNs) have demonstrated potential in predicting male fertility, with a few achieving remarkable accuracy. One ANN model achieved a maximum accuracy of 95% in predicting male fertility utilizing explainable AI [77].

  1. The authors discuss common computational methods such as ANNs and deep learning in the manuscript. In future work, the authors may consider the use of other computational methods in this field, such as graph representation learning (10.1109/JBHI. 2024.3357979).

Authors: We thank the reviewer for this suggestion. We added our consideration about this model adding the following sentence “Future studies on male infertility prediction may consider other computational methods, such as the exploitation of the graph contrastive clustering techniques [73] that are based on the use of deep neural networks for representation learning before clustering. This approach, for example, has the advantage of bringing similar or positive sample pairs closer and pushing dissimilar or negative sample pairs further away, thus going beyond the clustering methods of unsupervised machine learning algorithms”.

Reviewer 4 Report

Comments and Suggestions for Authors

Although, the research topic is really interesting but there is no novelty and research contribution. Apart from that this work is not presented in an appropriate manner.   The correction comments are aligned as follows, which may help to improve this article.

Abstract
Concern # 1: the abstract Reorganize the abstract to conclude:
(a)     The overall purpose of the paper and the research problems you investigated.
(b)     The basic design of the study.
(c)     Major findings or trends found as a result of the study.
(d)     A brief summary of your interpretations and conclusions.
** Instead of using distinct headings such as 'background and objectives', 'methods', 'results', and 'conclusion' in the abstract, these elements should dynamically emerge from the content itself.

Introduction
Concern # 1: The introduction should express and highlight the work, challenges, and solutions related to proposed research. However, I am not able to find any of novel or unique contribution.
Concern #2 The flow of introduction is not giving any appropriate path to reach any particular objective.
Concern #3 The statements should be validated with the appropriate references.
Related Work
Concern # 1: It should highlight the approach, limitations of the existing work related to proposed research.
Concern #2 All the relative works should be explained in a connective manner. At the end, the related work should reflect the objectives of proposed work.
Method and material
Concern#1 Need to introduce some new approaches or database to claim a technical contribution.
Concern #2 Need to explain the proposed methodology in appropriate manner.
Concern#3 A comprehensive and mathematical discussion is missing.
Results and Discussion
Concern #1 Use appropriate and technical performance metrices to evaluate the performance.
Concern#2 Need to technical explanation and justification of the proposed approach in detailed manner.
Concern#3 Need to involve and evaluate the proposed approach with the SOTA models.
Concern#4 Use suitable and important figures only avoid using unnecessary figures and tables.
Concern#5 It is clearly shown that the training data has imbalancing issue. So how it is used in the proposed work.
Conclusion
Concern # 1: Highlight your analysis and reflect only the important points for the whole paper.
*       Mention the benefits.
*       Mention the implication in the last of this section.
Concern # 2: For future work, try to include more than one direction.
** The paper's formatting does not align with the journal's guidelines, resulting in a subpar presentation. Authors are advised to seek assistance from professional and technical writing experts for improvement.

Comments on the Quality of English Language

Although, the research topic is really interesting but there is no novelty and research contribution. Apart from that this work is not presented in an appropriate manner.   The correction comments are aligned as follows, which may help to improve this article.

Abstract
Concern # 1: the abstract Reorganize the abstract to conclude:
(a)     The overall purpose of the paper and the research problems you investigated.
(b)     The basic design of the study.
(c)     Major findings or trends found as a result of the study.
(d)     A brief summary of your interpretations and conclusions.
** Instead of using distinct headings such as 'background and objectives', 'methods', 'results', and 'conclusion' in the abstract, these elements should dynamically emerge from the content itself.

Introduction
Concern # 1: The introduction should express and highlight the work, challenges, and solutions related to proposed research. However, I am not able to find any of novel or unique contribution.
Concern #2 The flow of introduction is not giving any appropriate path to reach any particular objective.
Concern #3 The statements should be validated with the appropriate references.
Related Work
Concern # 1: It should highlight the approach, limitations of the existing work related to proposed research.
Concern #2 All the relative works should be explained in a connective manner. At the end, the related work should reflect the objectives of proposed work.
Method and material
Concern#1 Need to introduce some new approaches or database to claim a technical contribution.
Concern #2 Need to explain the proposed methodology in appropriate manner.
Concern#3 A comprehensive and mathematical discussion is missing.
Results and Discussion
Concern #1 Use appropriate and technical performance metrices to evaluate the performance.
Concern#2 Need to technical explanation and justification of the proposed approach in detailed manner.
Concern#3 Need to involve and evaluate the proposed approach with the SOTA models.
Concern#4 Use suitable and important figures only avoid using unnecessary figures and tables.
Concern#5 It is clearly shown that the training data has imbalancing issue. So how it is used in the proposed work.
Conclusion
Concern # 1: Highlight your analysis and reflect only the important points for the whole paper.
*       Mention the benefits.
*       Mention the implication in the last of this section.
Concern # 2: For future work, try to include more than one direction.
** The paper's formatting does not align with the journal's guidelines, resulting in a subpar presentation. Authors are advised to seek assistance from professional and technical writing experts for improvement.

Author Response

We want to thank the Editor and the Reviewers for their positive general judgments about our manuscript and for their qualified comments that helped us to improve our work in its revised version. In the following our point-by-point responses with the changed parts of the manuscript are reported between apices and in italic style and highlighted in the revised version of the manuscript by word change tracking.

REVIEWER #4

Although, the research topic is really interesting but there is no novelty and research contribution. Apart from that this work is not presented in an appropriate manner.   The correction comments are aligned as follows, which may help to improve this article.

Authors: we thank the reviewer #4 for his overall judgment on our work. Please note that as directly expressed in the title, abstract and text, this paper is about a review of the existing literature. Therefore, although the reviewer's comments are of considerable importance, most of them are not possibly addressed.

Abstract
Concern # 1: the abstract Reorganize the abstract to conclude:
(a)     The overall purpose of the paper and the research problems you investigated.
(b)     The basic design of the study.
(c)     Major findings or trends found as a result of the study.
(d)     A brief summary of your interpretations and conclusions.
** Instead of using distinct headings such as 'background and objectives', 'methods', 'results', and 'conclusion' in the abstract, these elements should dynamically emerge from the content itself.

Authors: Thank you for these suggestions for improving our abstract. We have removed headings and to address the comments of the reviewer (taking also into account those of the other reviewers), we have modified the abstract as follows:

Male infertility is a relevant public health problem, but there is no systematic review of the different Machine learning (ML) models and their accuracy so far. The present review aims to comprehensively investigate the use of ML algorithms in predicting male infertility, also taking into account the accuracy of the used models. Particular attention will be paid to the use of Artificial Neural Networks (ANNs).

A comprehensive literature search was conducted in PubMed, Scopus, and Science Direct between July 15th and October 23rd, 2023, conducted under the Preferred Reporting Items for Systematic Reviews and Meta-Analyses (PRISMA) guidelines. We performed a quality assessment of the included studies using the recommended tools suggested for the type of study design adopted. We also made a screening of Risk of Bias (RoB) associated with the included studies.

43 relevant publications were included in this review, for a total of 40 different ML models detected. The studies included reported a good quality, even if RoB was not always good for all the types of studies.

The accuracy of ML models was greater than 88%. We found a few studies using ANN for male infertility prediction, reporting a mean accuracy of 84%.”

Introduction
Concern # 1: The introduction should express and highlight the work, challenges, and solutions related to proposed research. However, I am not able to find any of novel or unique contribution.

Authors: We thank the reviewer for his/her critical judgment. We have now stressed the novel aspects of our review, in particular, we have highlighted the gaps present in the literature that we wanted to fill up. We have now added the following paragraph:

Currently, in the scientific literature, there are not many review documents that have specifically addressed the use of machine learning models for predicting male infertility. A recent review [24] addressed the topic in a non-systematic and narrative way. The authors have drawn up examples of possible applications of ML algorithms in the hospital setting, providing indications on robots and mechanics in vogue for specialized analyses such as Advancing Computer-Assisted Semen Analyses (CASA), in the investigation of preoperative images before surgical sperm retrieval, for intra-operative advancement to identify genetic material at a microscopic level and sperm identification and analysis. This review, although concise and well-argued, has the limitation of being focused on assisted reproductive technologies without any specific reference to machine learning models. Another review [25] analysed the role of ML models exclusively for sperm selection before fertilization practices without considering other factors. Two other reviews have instead investigated azoospermia in patients with Klinefelter Syndrome [26] or in an all-encompassing way [27] about both male and female fertility, raising important consideration about the need for data augmentation, feature extraction, explainability, and the need to revisit the meaning of an effective system for fertility analysis.”.

Then we have also written:

“For this reason, it seems important to provide precise indications on what types of algorithms are currently used in the scientific panorama, being able to discriminate specific models and data on which they have been used. Among the multiple ML models currently available, Artificial Neural Networks (ANNs) are widely used in medical settings to improve the delivery of care at a reduced cost [28]. These models are particularly interesting since they are inspired by the neural organization of the human brain, which is modelled as an interconnection of nodes. In the context of male infertility, ANNs have played a crucial role in predicting sperm concentration [29]. This predictive ability is invaluable in assessing sperm quality, an essential determinant of male fertility.

(…) this review aims to fill this gap and contribute to the existing literature by critically evaluating the accuracy of ML in predicting male infertility and gaining insights about which model might be more efficient.”

Concern #2 The flow of introduction is not giving any appropriate path to reach any particular objective.

Authors: We thank the reviewer for this comment. According to all reviewers’ suggestions, we modified the introduction by editing sentences and adding references and paragraphs where it was requested as reported above in response to your previous comment.

[8] Cousineau, T. M., & Domar, A. D. (2007). Psychological impact of infertility. Best practice & research Clinical obstetrics & gynaecology21(2), 293-308.

[9] Skakkebaek, N. E., Lindahl-Jacobsen, R., Levine, H., Andersson, A. M., Jørgensen, N., Main, K. M., ... & Juul, A. (2022). Environmental factors in declining human fertility. Nature Reviews Endocrinology, 18(3), 139-157.

[12] O'Connor, J. C., & Chapin, R. E. (2003). Critical evaluation of observed adverse effects of endocrine active substances on reproduction and development, the immune system, and the nervous system. Pure and applied chemistry, 75(11-12), 2099-2123.

[14] Kumar, N., & Singh, A. K. (2022). Impact of environmental factors on human semen quality and male fertility: a narrative review. Environmental Sciences Europe, 34, 1-13.

[24] Diaz, P., Dullea, A., Chu, K. Y., Zizzo, J., Loloi, J., Reddy, R., … & Ramasamy, R. (2022). Future of male infertility evaluation and treatment: brief review of emerging technology. Urology169, 9-16.

[25] You, J. B., McCallum, C., Wang, Y., Riordon, J., Nosrati, R., & Sinton, D. (2021). Machine learning for sperm selection. Nature Reviews Urology18(7), 387-403.

[26] Krenz, H., Sansone, A., Fujarski, M., Krallmann, C., Zitzmann, M., Dugas, M., … & Gromoll, J. (2022). Machine learning based prediction models in male reproductive health: Development of a proof‐of‐concept model for Klinefelter Syndrome in azoospermic patients. Andrology10(3), 534-544.

[27] GhoshRoy, D., Alvi, P. A., & Santosh, K. C. (2023). AI Tools for Assessing Human Fertility Using Risk Factors: A State-of-the-Art Review. Journal of Medical Systems47(1), 91.

[28] Shahid, N., Rappon, T., & Berta, W. (2019). Applications of artificial neural networks in health care organizational decision-making: A scoping review. PloS one14(2), e0212356.

Concern #3 The statements should be validated with the appropriate references.

Authors: We added further references to our statements.

[8] Cousineau, T. M., & Domar, A. D. (2007). Psychological impact of infertility. Best practice & research Clinical obstetrics & gynaecology21(2), 293-308.

[9] Skakkebaek, N. E., Lindahl-Jacobsen, R., Levine, H., Andersson, A. M., Jørgensen, N., Main, K. M., ... & Juul, A. (2022). Environmental factors in declining human fertility. Nature Reviews Endocrinology, 18(3), 139-157.

[12] O'Connor, J. C., & Chapin, R. E. (2003). Critical evaluation of observed adverse effects of endocrine active substances on reproduction and development, the immune system, and the nervous system. Pure and applied chemistry, 75(11-12), 2099-2123.

[14] Kumar, N., & Singh, A. K. (2022). Impact of environmental factors on human semen quality and male fertility: a narrative review. Environmental Sciences Europe, 34, 1-13.

[24] Diaz, P., Dullea, A., Chu, K. Y., Zizzo, J., Loloi, J., Reddy, R., … & Ramasamy, R. (2022). Future of male infertility evaluation and treatment: brief review of emerging technology. Urology169, 9-16.

[25] You, J. B., McCallum, C., Wang, Y., Riordon, J., Nosrati, R., & Sinton, D. (2021). Machine learning for sperm selection. Nature Reviews Urology18(7), 387-403.

[26] Krenz, H., Sansone, A., Fujarski, M., Krallmann, C., Zitzmann, M., Dugas, M., … & Gromoll, J. (2022). Machine learning based prediction models in male reproductive health: Development of a proof‐of‐concept model for Klinefelter Syndrome in azoospermic patients. Andrology10(3), 534-544.

[27] GhoshRoy, D., Alvi, P. A., & Santosh, K. C. (2023). AI Tools for Assessing Human Fertility Using Risk Factors: A State-of-the-Art Review. Journal of Medical Systems47(1), 91.

[28] Shahid, N., Rappon, T., & Berta, W. (2019). Applications of artificial neural networks in health care organizational decision-making: A scoping review. PloS one14(2), e0212356.

Related Work
Concern # 1: It should highlight the approach, limitations of the existing work related to proposed research.
Concern #2 All the relative works should be explained in a connective manner. At the end, the related work should reflect the objectives of proposed work.

Authors: we thank the reviewer for this suggestion. As reported above, we think that also this comment has already been solved according to the reviewer’s suggestions.

Method and material
Concern#1 Need to introduce some new approaches or database to claim a technical contribution.

Authors: We thank the reviewer for having pointed out this aspect. Since it was not our aim to claim a technical contribution, we now stated it directly in the text. “Since the aim of this review was to make a comprehensive panoramic of ML models used to predict male infertility, we did not introduce any new approaches or databases to claim a technical contribution.”

We further stressed this aspect by accounting for it as a limitation of the current review which needs to be explored with an experimental design.

“Finally, considering that claiming a technical contribution was out of the scope of this review, we think that further research should address this point by introducing new approaches or databases”.

Concern #2 Need to explain the proposed methodology in appropriate manner.

Authors: Thank you for this comment. We think to have now clearly explained the methodology in the following paragraph:

“We used the PICO framework [30] was used for formulating the research questions reported in Table 1. The literature search was undertaken between August 2023 and October 2023 by two independent researchers (VS, DDB) using three databases (PubMed, Scopus, and ScienceDirect). Therefore, a search string was created as similar as possible for every database. The search strategy was conducted through the combination of Mesh, Tiab, and synonym terms for the general string of “Man, infertility, prediction”. In Appendix A we reported specifications about Mesh, Tiab, synonyms (table A1), and the specific strings search (table A2) adopted for each database. This review followed the Preferred Reporting Items for Systematic Reviews and Me-ta-Analyses (PRISMA) – Checklist [31].

All results from Pubmed, Scopus, and Science Direct searches were aggregated in an Excel sheet arranging the titles in alphabetical order. Since the aim of this review was not to make a comprehensive panoramic of ML models used to predict male infertility, we did not introduce any new approaches or databases to claim a technical contribution.

2.1. Inclusion criteria

After removing duplicates manually, the search results underwent a title and abstract screening, applying criteria for inclusion and exclusion of search results as follows:

-           Inclusion: Studies published in a peer-reviewed journal; any year of publication; all study designs; human male population.

-           Exclusion: Any language other than English; grey literature, letter to the editor or reviews; search results with content not directly relevant to the research question after; undergoing a title and abstract screening.; studies with different target populations (female, animals); articles with ambiguity in the context of male infertility in humans and machine learning; paper with no direct access to the full text.

The reference lists of the included studies were screened for further relevant publications. For a paper in which animal and male models are mutually included, we considered only data relating to human male infertility. 

2.2. Quality of the studies and Risk of Bias analysis

Two independent researchers analyzed the quality (VS) and the Risk of Bias (RoB) (DDB) of the included studies. The selected studies were grouped following the study design, and then a suitable tool for quality appraisal had to be chosen according to the study design. Retrospective and prospective studies were both assessed with the JBI Checklist for Case Series which consists of a rapid assessment based on the identification of ten specific information regarding the participants (from enrolment to the final stage of the experiment) and the procedure [32]. The remaining studies were assessed according to the EQUATOR guidelines [33]. Indeed, cohort, case-control, and cross-sectional studies were assessed through the STROBE guidelines for reporting observational studies. This tool is made of 22 items assessing in detail all the sections of the paper also including sub-items to further distinguish specific criteria of scoring according to the study design (cohort, case-control, cross-sectional) for the participants, the statistical methods, and the results sections. Studies using a multivariable prediction model were assessed through the Transparent Reporting of a multivariable prediction model for Individual Prognosis or Diagnosis (TRIPOD) checklist [34]. Similar to the STROBE, this tool is made up of 22 items including sub-items, thus collecting a scoring made up of 35 investigated domains.

Since these tools do not provide a cut-off scoring for further classification of the study quality, according to a previous study [35], we considered as good quality an assessed adherence ranging the 60-80%, computed as the scoring of the study on the total value of the tool. This normalization was applied to all the studies included in this review and allowed us to make a direct comparison of studies with different designs.

The RoB analysis was performed through the administration of the prediction model risk of bias assessment tool (PROBAST) [36]. This instrument is suitable for reviews of studies about clinical prediction models. The PROBAST checklist assesses the risk that arises from the methods used and the consequential applicability of the prediction model.

Concern#3 A comprehensive and mathematical discussion is missing.

Authors: The reviewer is right and we have added the following sentence to the limit section about our manuscript. In particular, we have written:
“Another limitation of our study is that a mathematical discussion of the ML algorithms used in the reviewed studies is missed. Although an interesting point of view, this aspect was excluded because it was out of the scope of this review and because it would need considerable work most suitable for a book or an article collection.”.

We have specified it because neither the Journal (Healthcare, which has readers more interested in health-related aspects than in computational aspects) nor the special issue to which we have submitted this manuscript is focused on mathematical details. 

Results and Discussion
Concern #1 Use appropriate and technical performance metrices to evaluate the performance.
Concern#2 Need to technical explanation and justification of the proposed approach in detailed manner.
Concern#3 Need to involve and evaluate the proposed approach with the SOTA models.

Authors: Thank you for these suggestions. About the concern #1 we have used the following validated approaches: PICO framework, PRISMA checklist, about the metrics to assess the papers we have used  JBI and TRIPOD checklists and EQUATOR and STROBE guidelines. We also used the PROBAST checklist to assess the risk of bias in the reviewed studies.

About the concern #2 we have provided a detailed description of the selected approaches writing:

“Two independent researchers analyzed the quality (VS) and the Risk of Bias (RoB) (DDB) of the included studies. The selected studies were grouped following the study design, and then a suitable tool for quality appraisal had to be chosen according to the study design. Retrospective and prospective studies were both assessed with the JBI Checklist for Case Series which consists of a rapid assessment based on the identification of ten specific information regarding the participants (from enrolment to the final stage of the experiment) and the procedure [32]. The remaining studies were assessed according to the EQUATOR guidelines [33]. Indeed, cohort, case-control, and cross-sectional studies were assessed through the STROBE guidelines for reporting observational studies. This tool is made of 22 items assessing in detail all the sections of the paper also including sub-items to further distinguish specific criteria of scoring according to the study design (cohort, case-control, cross-sectional) for the participants, the statistical methods, and the results sections. Studies using a multivariable prediction model were assessed through the Transparent Reporting of a multivariable prediction model for Individual Prognosis or Diagnosis (TRIPOD) checklist [34]. Similar to the STROBE, this tool is made up of 22 items including sub-items, thus collecting a scoring made up of 35 investigated domains.

Since these tools do not provide a cut-off scoring for further classification of the study quality, according to a previous study [35], we considered as good quality an assessed adherence ranging the 60-80%, computed as the scoring of the study on the total value of the tool. This normalization was applied to all the studies included in this review and allowed us to make a direct comparison of studies with different designs.

The RoB analysis was performed through the administration of the prediction model risk of bias assessment tool (PROBAST) [36]. This instrument is suitable for reviews of studies about clinical prediction models. The PROBAST checklist assesses the risk that arises from the methods used and the consequential applicability of the prediction model.

About Concern #3 We have deeply revised the State-Of-The-Art literature reporting the published algorithms for assessing male infertility. In the revised version of our manuscript, we have ameliorated also this aspect as detailed in our responses to the reviewers. We hope that the reviewer agrees with us appreciates all our efforts in ameliorating the manuscript according to his/her suggestions, otherwise, we need more detailed comments about specific parts of our manuscript.

Concern#4 Use suitable and important figures only avoid using unnecessary figures and tables.

Authors: Thank you for this general comment, We confirm that in the revised version of our manuscript only figures and tables presented in the main text of the paper are used to present results, data and information not reported elsewhere in the manuscript. Some tables that are useful, but not strictly necessary (but neither unnecessary) have been reported as Supplementary tables in the Appendices of the manuscript. We hope the reviewer appreciated this approach which we think is perfectly in line with his/her request.

Concern#5 It is clearly shown that the training data has imbalancing issue. So how it is used in the proposed work.

Authors: We thank the reviewer for this interesting comment. According to other reviewers’ suggestions, we added a paragraph in the Discussion section about this important concern.

“Indeed, in the included studies we found huge differences in input information used to build models. As we reported in tables ranging from 1-4, the data sources used for building each model were very different, including demographical data, RNA, quality of the embryo, sperm retrieval, hormonal data, and others. This variety, in addition to the fact that in many studies multiple variables were used as a combination matrix, contributed to limiting possible direct comparison among models.”

Conclusion
Concern # 1: Highlight your analysis and reflect only the important points for the whole paper.
*       Mention the benefits.
*       Mention the implication in the last of this section.
Concern # 2: For future work, try to include more than one direction.

Authors: Authors: We thank the reviewer for this interesting comment. About benefits and implications, in the revised version of the manuscript there is now written:

Models based on artificial neural networks (ANN) have demonstrated potential in predicting male fertility, with a few achieving remarkable accuracy. One ANN model achieved a maximum accuracy of 95% in predicting male fertility utilizing explainable AI [77]. Deep Learning techniques, a subset of ML, have been employed to evaluate sperm motility and morphology [44,45,68]. Unlike traditional methods, CNNs excel at processing visual data, making them particularly well-suited for tasks involving embryo morphology, and have also demonstrated effectiveness in assessing spermatozoid motility [44,45,47,53]. Certain investigations might employ a combination of ML, ANN, and DL techniques for a comprehensive analysis, harnessing the strengths of each.

Choosing one model based just on the highest accuracy reported might be an inappropriate decision since it is crucial to consider the nature of the data, the complexity of the prediction task, and the interpretability of the results when selecting the most suitable model. We conclude that the use of ML models for the identification of risk factors and prediction of male infertility has the potential to assist personalized medicine in achieving its goal of creating optimally tailored diagnostic, preventive, and therapeutic measures”.

Then, according to this and other reviewers’ suggestion, we included the  possible future directions:

“Future studies on male infertility prediction may consider other computational methods, such as the exploitation of the graph contrastive clustering techniques [82] that are based on the use of deep neural networks for representation learning before clustering. This approach, for example, has the advantage of bringing similar or positive sample pairs closer and pushing dissimilar or negative sample pairs further away, thus going beyond the clustering methods of unsupervised machine learning algorithms”.

** The paper's formatting does not align with the journal's guidelines, resulting in a subpar presentation. Authors are advised to seek assistance from professional and technical writing experts for improvement.

Authors: we confirm that we used the template of the Journal to write the paper and that therefore it does align with the journal guidelines.

Round 2

Reviewer 3 Report

Comments and Suggestions for Authors

All of my concerns were addressed in this revision.

Comments on the Quality of English Language

The quality of English language is acceptable.

Author Response

We thank the reviewer for his/her positive final judgement.

Reviewer 4 Report

Comments and Suggestions for Authors

Minor revisions are required in the article before final publication.  Please address these revisions and resubmit your article for final acceptance.

1. Abstract: Reduce the abstract in 350 words 

2. Introduction: Clearly state the research problem and objective early in the introduction. The introduction provides a good overview, but it could be more engaging. Start with a compelling hook or a real-world scenario that demonstrates the importance of IoT.

3. Review of Literaure: Include the some recent works like: COVID-19 Detection using Hybrid CNN-RNN Architecture with Transfer Learning from X-Rays; Unveiling the prevalence and risk factors of early stage postpartum depression: a hybrid deep learning approach; Application of machine learning for cardiovascular disease risk prediction 

4. Methodology: Include the rationale behind choosing this methodology and how it addresses the predictive Maintenance issues.

5. Describe the steps and procedures in a clear and logical order.

6. In results and discussion: Avoid restating results in the discussion section; focus on the implications and significance of the findings.

7. Figures and Tables: Make sure each figure and table is appropriately labeled, and their titles and captions should be self-explanatory.

8. Conclusion: Avoid introducing new information in the conclusion section.

9. Future Scope: Add some futuristic aspects for this domain.

Author Response

We thank the Reviewer for his/her positive general judgments about our manuscript and for his/her qualified comments that helped us to improve our work in its revised version. In the following our point-by-point responses with the changed parts of the manuscript are reported here between apices and in italic style and highlighted in the revised version of the manuscript by word change tracking.

  1. Abstract: Reduce the abstract in 350 words 

Authors: we thank the reviewer for this comment. We confirm that the abstract does not exceed 350 words.

  1. Introduction: Clearly state the research problem and objective early in the introduction. The introduction provides a good overview, but it could be more engaging. Start with a compelling hook or a real-world scenario that demonstrates the importance of IoT.

Authors: we thank the reviewer for his/her positive judgment on the modification provided in the manuscript. The introduction starts with the exposition of the current problem of male infertility “Approximately 30% of infertility cases are attributed to male factors [4].  Recent meta-analysis studies show that male factors are present in 20–70 per cent of infertility cases [5]. These findings are significantly broader than previously reported [4,5]. However, the wide range of male infertility in meta-analysis studies [6] may not reflect the prevalence of this complication in all parts of the world because of reasons such as the lack of rigorous statistical methods that include bias, heterogeneity in data collection, and cultural constraints [6]. Indeed, the inability to conceive a child can be emotionally taxing, leading to feelings of inadequacy and Men may contemplate their masculinity, as societal standards frequently associate potency with manliness [..] Recent studies addressed the investigation of genetic abnormalities [10], such as chromosomal abnormalities and gene mutations [11], as factors affecting sperm production and function [10]. Furthermore, the role of hormonal imbalances, such as low levels of testosterone or high levels of prolactin, in disrupting the normal functioning of the reproductive system and sperm production has been investigated [12]. Indeed, lifestyle factors, including smoking, excessive alcohol consumption, drug use, and exposure to environmental toxins, can also contribute to male infertility [13, 14]. Recently there has been an increasing interest in chemical exposure as a cause of male infertility [14], which may likely play a substantial role in the sperm count trend. [..]

and continues with the possibility of using machine learning-based models to predict the factors associated with this issue:

“Given the complexity of the topic, it is essential to have predictive models that can provide accurate information on the factors that influence male infertility. Traditionally, the total motile sperm count (volume × concentration × motility) has been used as the most predictive factor in determining fertility compared to volume, concentration, and motility individually [17]. On the other hand, a recent study [18] reported that semen analysis (sperm quality) is a fundamental method for evaluating male fertility, including the analysis of sperm count, motility, morphology, and volume. Therefore, abnormalities in these parameters can indicate male infertility. Since many factors have been investigated as potential risk factors related to male infertility [5,7,9-17], it is important to identify statistical models that can accurately predict how much each single risk factor weighs on the onset of male infertility.

The introduction of Artificial Intelligence (AI) into the healthcare field promoted a significant change in the approach of medical practitioners to diagnosing, treating, and anticipating health ailments. Indeed, the use of AI, which is powered by machine learning algorithms, data, and computational capabilities, allows for the analysis of large datasets with impressive speed [18]. This transformative technology can predict illnesses, categorize patient information, and offer tailored treatment recommendations. Predictive analytics in medicine leverages AI to forecast patient outcomes, identify potential health risks, and optimize care pathways using algorithms such as Random Forest (RF) Classifier [19,20]. Exploiting different types of information AI models can forecast the commencement and advancement of diseases, such as in the study of heart disease [21], cancer [22], and diabetes [23] in which AI model has already proven its efficiency in facilitating timely intervention and tailored healthcare strategies.

Machine learning (ML), artificial neural networks (ANNs), and deep learning have emerged as feeble tools, stagnating our approach to comprehending and addressing male reproductive health. These tools have been employed to scrutinize extensive datasets, thereby assisting in the identification of pivotal factors that influence fertility outcomes.”

Following all the reviewers’ suggestion, we specified in the manuscript that “the aim of this review was to make a comprehensive panoramic of ML models used to predict male infertility, we did not introduce any new approaches or databases to claim a technical contribution” and also “to support those who are approaching the use of machine learning in the medical field for the first time in choosing the most suitable model, based on the accuracy levels already established by previous research.” For this reason, we cannot introduce the importance of internet of things in the Introduction section, because we fear that the clinical audience to which this review is aimed may not benefit from such technical information.

  1. Review of Literaure: Include the some recent works like: COVID-19 Detection using Hybrid CNN-RNN Architecture with Transfer Learning from X-Rays; Unveiling the prevalence and risk factors of early stage postpartum depression: a hybrid deep learning approach; Application of machine learning for cardiovascular disease risk prediction 

Authors: We thank the reviewer for this suggestion. We added some of them in the literature review.

  1. Methodology: Include the rationale behind choosing this methodology and how it addresses the predictive Maintenance issues.

Authors: We thank the reviewer for this comment. Since this paper is a review, we followed the recommended guidelines reported in the following sentences:

We used the PICO framework [30] was used for formulating the research questions [..]

This review followed the Preferred Reporting Items for Systematic Reviews and Meta-Analyses (PRISMA) – Checklist [31]. [..]

The selected studies were grouped following the study design, and then a suitable tool for quality appraisal had to be chosen according to the study design. [..]

The RoB analysis was performed through the administration of the prediction model risk of bias assessment tool (PROBAST) [36]. This instrument is suitable for reviews of studies about clinical prediction models. The PROBAST checklist assesses the risk that arises from the methods used and the consequential applicability of the prediction model”.

Following the reviewer’s previous suggestion, we specified that this review is not to be intended as a technical contribution. Although the issue of the predictive maintenance is an important point in ML models, this aspect cannot be addressed in a review paper.

  1. Describe the steps and procedures in a clear and logical order.

Authors: we did not understand what this sentence refers to.

  1. In results and discussion: Avoid restating results in the discussion section; focus on the implications and significance of the findings.

Authors: We thank the reviewer for this valuable comment. We checked the discussion and we have edited the sentence in which a result was reported.

  1. Figures and Tables: Make sure each figure and table is appropriately labeled, and their titles and captions should be self-explanatory.

Authors: We confirm that each figure and table are appropriately labelled and that their titles and captions are self-explanatory.

  1. Conclusion: Avoid introducing new information in the conclusion section.

Authors: We confirm that the conclusion does not introduce new information.

  1. Future Scope: Add some futuristic aspects for this domain.

Authors: We thank the reviewer for this comment. In accordance with all the reviewers, this section was modified as follows: “Future studies on male infertility prediction may consider other computational methods, such as the exploitation of the graph contrastive clustering techniques [82] that are based on the use of deep neural networks for representation learning before clustering. This approach, for example, has the advantage of bringing similar or positive sample pairs closer and pushing dissimilar or negative sample pairs further away, thus going beyond the clustering methods of unsupervised machine learning algorithms.”